# Evidence of Prolonged Monitoring of Trauma Patients Admitted via Trauma Resuscitation Unit without Primary Proof of Severe Injuries

**DOI:** 10.3390/jcm9082516

**Published:** 2020-08-04

**Authors:** Martin Heinrich, Matthias Lany, Lydia Anastasopoulou, Christoph Biehl, Gabor Szalay, Florian Brenck, Christian Heiss

**Affiliations:** 1Department of Trauma, Hand and Reconstructive Surgery, University Hospital Giessen, 35392 Giessen, Germany; Matthias.Lany@chiru.med.uni-giessen.de (M.L.); Lydia.Anastasopoulou@chiru.med.uni-giessen.de (L.A.); Christoph.Biehl@chiru.med.uni-giessen.de (C.B.); Gabor.Szalay@chiru.med.uni-giessen.de (G.S.); Christian.Heiss@chiru.med.uni-giessen.de (C.H.); 2Experimental Trauma Surgery, Justus-Liebig-University of Giessen, 35392 Giessen, Germany; 3Department of Anesthesiology, Intensive Care Medicine and Pain Therapy, University Hospital Giessen, 35392 Giessen, Germany; Florian.Brenck@chiru.med.uni-giessen.de

**Keywords:** trauma resuscitation unit, emergency medicine, injury severity, abbreviated injury scale, injury severity score

## Abstract

Introductio: Although management of severely injured patients in the Trauma Resuscitation Unit (TRU) follows evidence-based guidelines, algorithms for treatment of the slightly injured are limited. Methods: All trauma patients in a period of eight months in a Level I trauma center were followed. Retrospective analysis was performed only in patients ≥18 years with primary TRU admission, Abbreviated Injury Scale (AIS) ≤ 1, Maximum Abbreviated Injury Scale (MAIS) ≤ 1 and Injury Severity Score (ISS) ≤3 after treatment completion and ≥24 h monitoring in the units. Cochran’s Q-test was used for the statistical evaluation of AIS and ISS changes in units. Results: One hundred and twelve patients were enrolled in the study. Twenty-one patients (18.75%) reported new complaints after treatment completion in the TRU. AIS rose from the Intermediate Care Unit (IMC) to Normal Care Unit (NCU) 6.2% and ISS 6.9%. MAIS did not increase >2, and no intervention was necessary for any patient. No correlation was found between computed tomography (CT) diagnostics in TRU and AIS change. Conclusions: The data suggest that AIS, MAIS and ISS did not increase significantly in patients without a severe injury during inpatient treatment, regardless of the type of CT diagnostics performed in the TRU, suggesting that monitoring of these patients may be unnecessary.

## 1. Introduction

Acute care of trauma patients in Germany, Austria, Switzerland, the Netherlands, Belgium and Luxembourg is ensured by local, regional and supraregional trauma centers (level I-III) according to the TraumaNetwork DGU^®^ initiative.

TraumaNetwork DGU^®^ initiative has enabled the German Society for Trauma Surgery (DGU) to establish first-class nationwide care for the severely injured. Clinics and University hospitals in order to provide high-quality patient care in trauma centers in addition to specialized medical requirements, also require specific spatial and material resources [1,2]. According to the guidelines of the DGU and the “Association of the Scientific Medical Societies in Germany” specific criteria must be fulfilled for the activation of the Trauma Resuscitation Unit (TRU). These criteria are divided into three groups: “disturbance of vital signs”, “detected injuries” and “mechanism of the accident or accident constellation” [2,3]. However, based on the last criterion, it has been observed that in Germany that many slightly injured patients are admitted to the TRU, yet only one in five are severely injured with an Injury Severity Score (ISS) ≥ 16 [4,5].

In 2019 the German Society for Trauma Surgery published the “Whitebook Medical Care of the Severely Injured” 3rd edition, which provides guidelines for the clinical and diagnostic steps, as well as for the first operative phase of the “Trauma Resuscitation Unit treatment” [2,6,7]. However, until now, there are no recommendations for the duration of treatment and whether further monitoring is necessary for the slightly injured trauma patients, who have suffered a dangerous trauma and who according to the TRU criteria must be treated in the TRU.

In addition, it has been shown that the intake of mind-altering substances (e.g drugs, alcohol or medication) in combination with a dangerous trauma, even if the patient is slightly injured, leads to the monitoring of the patient [8]. In individual and defined cases, if the patient has not suffered a severe injury, after implementation of the standardized Trauma Resuscitation Unit treatment, and if the occurrence of possible further complications is excluded, prolonged monitoring appears not to be necessary [9,10,11]. However, relevant injuries (such as cerebral contusion, occipital skull fracture or pneumothorax) have been initially overlooked and were only subsequently detected [8].

This might be the reason, that it has been reported in the literature that many patients with minor trauma are admitted to the Intermediate Care Unit (IMC) for monitoring without any comprehensible medical reason [12].

A national or international study that considers the entire cohort of patients that have been treated in the TRU and were initially diagnosed as slightly injured does not exist.

In the present study, it is therefore investigated whether (1) inpatients surveillance after the interdisciplinary TRU diagnostic is necessary so that injuries with an Abbreviated Injury Scale (AIS) ≥2 are not overlooked; and (2) whether the inquired experience of the trauma team along with the improved computed tomography (CT) imaging makes the inpatient monitoring unnecessary.

## 2. Materials and Methods

### 2.1. University Hospital of Giessen and Data Acquisition

The University Hospital of Giessen, located in the middle of Germany, is a National Trauma Center (Level I) of the DGU. In 2018, 370 patients were admitted with moderate to severe injuries, which was the greatest number of patients registered by the TraumaRegister DGU^®^ in a single hospital throughout Germany at that time. In addition, a total of about 1000 trauma patients are treated annually in the TRU [13].

When treating trauma patients, the procedures of the TRU-algorithm at the University Hospital of Giessen strictly adhere to the Advanced Trauma Life Support (ATLS) protocol. More specifically, patient care is carried out by a defined TRU team, which consists of five attending physicians from the fields of trauma surgery, visceral surgery, anesthesia, neurosurgery and radiology. While the diagnostic steps of the primary survey are strictly defined in the treatment protocol and are identical for all patients, diagnostic imaging in the TRU is individually adapted to each trauma patient by the trauma team.

After admission in the TRU, regardless of the type of injuries, trauma patients are monitored for 24 h, initially in the Intermediate Care Unit (IMC). They are then transferred to the Normal Care Unit (NCU) until their discharge from the hospital. All patient documentation is carried out promptly in the electronic patient file. “MEONA” software from Meona GmbH, Freiburg, Germany is used at NCU and “icudata” from IMESO-IT GmbH, Giessen, Germany is used at IMC, both are used in the TRU.

### 2.2. Inclusion and Exclusion Criteria

Ethical approval was given by the Justus-Liebig-University Giessen’s ethics committee (approval number AZ 67/20). The study included during the eight month observation period between April and November 2019 data from 112 adult (≥18 years) trauma patients admitted to the TRU at the University Hospital of Giessen, Department of Trauma, Hand and Reconstructive Surgery, who had Maximum Abbreviated Injury Scale (MAIS) ≤1, Injury Severity Score (ISS) ≤3 and had been monitored at IMC or NCU for 24 h.

Children, as well as patients with less than 24-h monitoring or with severe injuries (MAIS ≥2) or patients with incomplete data files, were excluded from the study.

### 2.3. Definitions

All 112 patients injuries were rated using the AIS (AIS Version 2005/Update 2008, Association for the Advancement of Automotive Medicine, Barrington, IL), using a scale from 0 to 6 according to injury severity (Table 1) [14]. From the AIS the MAIS and ISS were derived [15]. Evaluation of injuries took place retrospectively, based on the electronic patient record at the end of the TRU management, the IMC-observation and at the end of hospitalization at NCU.

In the case of changes at the AIS during hospitalization, a consultant evaluated the consequences, and if necessary, advised for a change in therapy. If the patients reported new symptoms, the current location, IMC or NCU, was also noted.

Furthermore, additional parameters as gender, age and the course of the accident (“motor vehicle accident”, “motorcycle/bicycle accident”, “high and low falls”, “pedestrian vs. vehicle crash”, “corporal violence”, “others”) were included. The performance of a head and body trunk CT, only head or body trunk CT or no CT during the initial TRU assessment was noted. The respective medical units extended diagnostics that were performed were also recorded.

### 2.4. Statistical Analysis

For statistical assessment of the changes in the AIS and ISS a binary form of the two scores was encoded (AIS: =0 as 0 and >0 as 1; ISS: ≤3 as 0 and >3 as 1). Cochran´s Q tests were used for ISS and AIS variables. For the recoded data, the number and percentage of persons with an AIS/ISS = 1 in the respective hospital unit and the percentage of persons for whom the value has changed from TRU to IMC or IMC to NCU were also shown. Risk assessment and risk differences were also calculated. The Chi^2^-test was used to determine whether AIS changes depended on CT diagnostics carried out in the TRU.

Statistical analysis was performed using SPSS (Version 23, IBM Inc., Armonk, NY, USA), and the level of significance was 5%.

The total length of stay (LOS, in hours) in relation to different variables was also analyzed. A minimum LOS of 24 h was mandatory for all patients. Thus, this was defined as the null by subtracting 24 from all values. The explanatory variables were changed (three categories: “no change”, “new symptoms” and “AIS change”—with the first one used as a reference category for dummy coding to which the others were compared), age (in years), gender (men as reference category), and extended diagnostics (in this sample, people received extended diagnostics once at maximum if at all; thus, not receiving extended diagnostics was simply chosen as the reference category).

A generalized linear regression model was applied due to the right-skewed distribution of the variable “length of stay”. A gamma distribution was chosen in combination with the natural logarithm as the link function between mean length of stay and the explanatory variables. Then the regression coefficients were back-transformed using the exponential function. Hence, they represent a multiplicative change in length of stay. For all analyses, a significance level of 5% was chosen. Statistical analysis was carried out using R 3.6.3. [16].

Data are presented as mean ± standard deviation (mean ± SD). In the case of asymmetrical distribution, the median was also calculated.

## 3. Results

### 3.1. Demographic Data

The study included 112 (19.55%) out of the 573 trauma patients admitted in the TRU of the University Hospital of Giessen during the eight month observation period between April and November 2019 who fulfilled the inclusion criteria (≥18 years, primary TRU admission, AIS ≤ 1, MAIS ≤ 1 and ISS ≤ 3 after TRU treatment completion, ≥24 h monitoring in the units). Participants’ mean age was 41.18 ± 19.63 years; 69 (62%) were males and 43 (38%) females (Table 2).

The car accident was the reason for admission in 62 cases (55.36 %). Other accidents included falls in 20 cases (17.86%), accidents between cars and pedestrians or cyclists in 12 cases (10.71%), falls from motorcycles or bicycles in nine cases (8.04%), physical abuse in five cases (4.46%), and other types of accidents (circular saw and pinch point injuries) in four cases (3.57%) (Table 3).

### 3.2. Average ISS, AIS and MAIS

All patients admitted in the TRU had an average ISS of 1.45 after treatment. The average score increased to 1.47 in the IMC and to 1.55 in the NCU. An overall increase of 6.9% from the TRU to NCU was observed. The average AIS of all regions rose from an initial 1.45 in the TRU to 1.47 in IMC and to 1.54 in NCU, resulting in an overall increase of 6.2%. The MAIS was 1 in the TRU and IMC and 2 in the NCU. Out of the 112 patients, a total of four (3.57%) were completely unharmed, and therefore, had an AIS and ISS of 0. These scores did not change during inpatient hospitalization (Table 4).

The frequency distribution of the individual ISS-values throughout the different units showed that ISS in the IMC increased one point in only two patients from the initial ISS in the TRU. In six patients, the ISS change occurred in the NCU, and in one of these patients, the ISS rose to five (Table 5).

Figure 1 shows, that there was no significant change of the ISS during patients’ hospitalization in the different units. Means and medians were almost at the same level, suggesting that ISS values, and thus, the overall severity of injuries did not change significantly in the different units. Only in NCU the ISS mean rose slightly (ISS 1.55) which can be explained by an individual outlier (Figure 1).

### 3.3. New Complaints, New Diagnoses and AIS/ISS Changes

A total of 21 patients (18.75%) after completion of TRU treatment, complained about new symptoms, such as pain, abnormal sensations, and in one case, visual disturbances, during the course of their inpatient stay. Nine of these patients reported new symptoms in the IMC, while 10 only reported new symptoms in the NCU. Two of these patients reported new complaints on both IMC and NCU (Table 6). A new diagnosis was made in 9/112 patients (8.04%) that led to an AIS change. Eight of these patients (7.14%) had an AIS change, as well as an ISS change. In one patient, the MAIS rose from 1 to 2 (0.89%), and thus, the ISS to 5. In all of the rest patients the MAIS remained at 1 and ISS ≤ 3. A new diagnosis was made in IMC in 2/9 patients (22.2%) and in the other seven (77.8%) patients in the NCU.

Evaluation of the changes in AIS and ISS values during the course of inpatient stay showed no significant change in AIS values for any region of the body (AIS HN: *p* = 0.368; AIS F: *p* = 0.223; AIS C: *p* = 0.368; AIS AbP: *p* = 0.368; AIS ExP *p* = 0.135; AIS E: *p* = 0.223) and also no significant ISS increase (*p* = 0.368) (Table 7). Data analysis showed that the risk difference for an AIS increase by 1 in the extremities during an inpatient stay in NCU was 1.79% (*p* = 0.135). Table 7 shows the distribution of AIS ≥ 1 throughout the different units, as well as the probability of change (RD = Risk Difference).

### 3.4. TRU Computed Tomography (CT) and AIS Change

No correlation was found between CT diagnostics performed in the TRU and change in AIS during inpatient’s hospitalization (*p* = 0.542). Each of the nine patients, who had an AIS change after completion of TRU treatment, had received a CT diagnostic in the TRU. Eight of them (88.9%) received a trauma CT (head and body-trunk), and one patient only received a CT head. Of all patients who received a CT body trunk (80/112), eight (10%) had an AIS change despite the extensive TRU CT diagnostics. Only head CT diagnostic was performed in the TRU in 25/112 patients and only one of them, received during the hospitalization in the NCU X-rays of the shoulder, which led to an AIS change. The head CT examination that was carried out in the TRU had no influence on this change (Table 8).

### 3.5. Total Length of Stay (LOS) and Factors Influencing the Length of Stay

The average total LOS (TRU+IMC+NCU) was 53.57 h with a standard deviation of 42.23 h. Most hospitalization time was spent on the NCU, while the time spent on the IMC only accounted for 33.28% of total LOS. As expected, the length of stay at the TRU was short—with an average total of 1.08h (Table 9).

It was also shown, that the average length of stay of patients, who had reported new symptoms without any change in AIS was 65%, and of those who had changes in AIS was 85% of the length of stay of patients who had neither reported new symptoms nor had changes in AIS (*p* = 0.350 and *p* = 0.770) (Table 10, Figure 2).

The average total length of stay of female patients was shorter compared to male patients (*p* = 0.481).

The data further showed that for every one year increase in age, there was an increase in the length of stay by a factor of 1.01 (*p* = 0.064) (Figure 3). The necessity of extended diagnostics in the course of inpatients’ stay led to a statistically significant increase of the total length of stay by a factor of 3.61 (*p* = 0.023).

## 4. Discussion

The aim of this study was to examine whether monitoring of patients with only minor injuries or no injuries at all after completion of TRU treatment is beneficial in order to identify new relevant injuries during hospitalization. The authors retrospectively analyzed the data of 573 patients that were admitted in the TRU. One hundred and twelve of them had ISS ≤ 3 (MAIS ≤ 1) and were included in the study. This corresponds to 19.55% of all primary TRU patients of Level I trauma center in Giessen, Germany, in the examination period 2019 (April-November). In comparison to the annual report of the Trauma-Register DGU^®^ in 2019, the overall proportion of patients, who were only slightly injured or not injured at all was 13%. Since the slightly injured patients are only being recorded in the Trauma-Register without any further analysis of their data, it can be assumed that most of these patients do not enter the register, and therefore, they are not fully reflected in the statistics [17,18]. Marzi showed an increase of patient admission to TRU of 70 % from 2012 to 2016, although the number of patients with an ISS ≥ 16 or a MAIS ≥ 3 is slightly constant over the years [4]. We made a similar observation in our center and saw this as a reason for the high number of patients with MAIS ≤ 1.

The mean age of these study population was 41 ± 19 years in accordance with the literature (30–50 years). The large fluctuations in mean age can be explained by the differences in the inclusion and exclusion criteria between the various studies [8,11,19]. More specifically in the study of Lansink et al. in 2004 mean age was 30 years, but patients with known coronary artery disease, anticoagulant medication, blood clotting disorder and patients admitted in the ICU were excluded from the study [11]. As these criteria mostly apply to older patients, mean age was therefore smaller. The proportion of male participants in the present study was 62%, which is comparable to other study populations (66–70%) [8,10,11,18,19,20].

Considering our demographic data, the mechanisms of accidents in this study are comparable with the ones reported in previous studies [8,10,11,19], with the exception of Salim et al., who reported a slightly higher incidence of pedestrian struck-accidents (25.9%) [20].

The severity of the injury was assessed using the internationally recognized Abbreviated Injury Scale (AIS), which was first introduced in 1969. The AIS-Codebook, first published in 1979, has been since then continuously revised. The AIS^©^ 2005 Update 2008 was used for this study [14,21].

The AIS is the basis for calculating the Injury Severity Score (ISS) of a multiply injured patient; a scoring system which was first described in 1974 by Baker et al. [22,23]. In addition to the AIS and ISS, the Maximum Abbreviated Injury Scale (MAIS) was also determined, which gives the maximum AIS value of each patient and estimates the severity of an injury [17].

We observed that after completion of TRU diagnostics, mean AIS for all body regions and ISS were 1.45 and as none of our trauma patients had MAIS > 1 they all belonged to the category of slightly injured patients [14,22,23,24].

It was shown that AIS, ISS and MAIS did not change significantly during an inpatient hospital stay. Only in one case, AIS increased to 2 and ISS to 5, which led according to Baker et al. (1976) to an insignificant increase in mortality [23]. This delayed diagnosed injury was a sternal fracture, which was already demonstrated in the initial TRU imaging, but was not documented among the diagnoses in the electronic patient’s file.

There are only a few studies with a comparable population as this present study, as most clinical studies focus on missed injuries in severely injured patients. In our study, ISS increased by 6.9% during inpatient stay, while AIS increased by 6.2%. New diagnosis in the present study was made in 9/112 patients (8%). Four of the newly diagnosed injuries were classified as relevant but did not lead to a change of further treatment regimen. Zamboni et al. reported 7.6% newly diagnosed injuries in the course of inpatient stay, whereby at this study, the diagnoses were only mentioned without a calculation of AIS or ISS [25]. Moreover, Stephan et al. reported new diagnoses in 35/630 patients (5.56%) during a 23-h patient surveillance period, of which 14 (2.22%) were relevant [8].

Lansink et al. evaluated the necessity of clinical observation of high-energy trauma patients without significant injury. In this study, the patients’ ISS was higher than ours (1.37–3.09), and it did not increase during inpatients’ hospitalization. The authors attributed their findings, to the high-quality examination in the TRU based on ATLS guidelines [11]. The authors of the present study also believe that the reason for not having many new diagnosed (previously overlooked) injuries in the course of inpatients’ stay, is the application of the standardized ATLS-based TRU treatment in Giessen, whereby the comparison between the two studies, due to the extensive exclusion criteria in Lansink et al. study, is restricted [11].

However, it is known, that there is a significant association between the severity of injury and occurrence of overlooked injuries, as in slightly injured patients, who were admitted in the TRU only a few injuries were subsequently detected [26]. In the current study, a large number of delayed diagnoses was not expected. However, relative studies report a widespread of incidence (1.3–39%) of delayed diagnosed injuries, and this can be attributed to the fact that each study has a different design, as well as different definitions for an overlooked injury. Pfeifer et al. 2008 reported that 15–22.3% of the subsequently diagnosed injuries were classified as clinical significant [26]. Regarding our discovered diagnoses we showed that in our study population, 44.44% of them were in extremities. This is in line with relative studies from the literature, in which the proportion of the overlooked extremity injuries is between 33–60% [27,28,29].

In addition, our finding that 3.57% of the patients had no injuries both in the TRU and in their further hospitalization is consistent with relative German literature [5].

Furthermore, our data showed that AIS and ISS, regardless of TRU diagnostics, did not significantly increase in the course of inpatient stay. It was only after the development of multislice CT that a whole-body CT was possible, and this was first described as a diagnostic tool for seriously injured patients in 1997 [30,31,32]. Since then, performing a whole-body computed tomography (CT) in the seriously injured patients in the TRU is recommended according to the German guidelines as “Grade of Recommendation A”, as among others, it increases the probability of survival of the severely injured patients [3,33,34,35] While the retrospective data in the literature appear to be clear considering the seriously injured patients, there are no recommendations regarding the indication for performing a whole-body CT or if necessary, a focused CT. Only the parameters “disturbed vital signs”, a “relevant mechanism of injury” and the presence of “at least two injured regions” can be used as “Grade of Recommendation B” [3,36,37].

An extensive imaging diagnostic alone does not appear to be required in order to discharge patients after TRU treatment or transferring them to the NCU. Kendall et al. defined some criteria for low risk of overlooking abdominal injuries after blunt abdominal trauma, which would allow the patient to be discharged from the emergency department (ED) without having a CT scan of the abdomen. The authors showed a low risk in the absence of intoxication, prehospital or ED settled hypotension, tachycardia in the ED, abdominal pain or tension, gross hematuria and distracting injury [10]. Similar results demonstrated Nagy et al. and Livingston et al. regarding head trauma patients with a Glasgow Coma Score (GCS) 15 and a negative head CT result. Both authors considered, in this case, an inpatient admission for further monitoring to be unjustified [38,39].

For the isolated high-energy trauma, Lansink et al. suggested that there is no evidence for inpatient monitoring if an ATLS-based TRU treatment has been previously carried out. However, many groups of patients were excluded from this study [11]. This observation by Lansink et al. is also in accordance with our results for all the TRU patients without a significant injury after completion of TRU diagnostics. The insignificant increase (*p* = 0.368) in the ISS, which is presented in the current study is a further indication that the ATLS-based TRU management increases the quality of treatment.

Another reason for a large number of slightly injured patients is the increasing use of the TRU in the last years leading to an increase in overtriage, and thus, to the admission of uninjured or slightly injured patients, whereby an overtriage of 25–35% appears to be necessary [4,40,41]. The high number of TRU activations after a specific mechanism of accident also play a decisive role here [5].

We demonstrated that there were no relevant changes in the AIS and ISS in the course of inpatient stay, regardless of how long the patient was hospitalized for. The average total length of stay in the current study was 2.23 days, which coincides with Stefan et al., who reported a total length of stay of three days (+/−2 days) [8]. The results of the present study were also comparable with other relative studies, in which patients were monitored between 8 and >24 h in an NCU, a clinical decision unit and an IMC [8,9,10,11,19].

## 5. Conclusions

The results of the present study lead to the conclusion that an individualized extended diagnostics using focused CT or whole-body CT combined with a specialized TRU medical team and an ATLS-based TRU treatment in a Level I trauma center is a good algorithm in order not to overlook relevant injuries in the TRU. Hence, an inpatient monitoring in the IMC for patients with an AIS ≤ 1 or an ISS ≤ 3 after completion of the TRU treatment is unnecessary.

In conclusion, our results show that patients who were diagnosed with no injuries or only minimal injuries in the TRU with an ISS ≤ 3 do not need to be monitored in the IMC, regardless of whether they have received in the TRU a trauma CT or only a basic diagnostic procedure according to the ATLS-standards, as clinical examination, extended focused assessment with sonography in trauma (eFAST) and X-rays.

The authors, therefore, suggest that the use of AIS/ISS, which is until now only used retrospectively, could be a good tool for deciding whether the monitoring of a slightly injured patient is necessary, or whether the patient after receiving, for example, a scheduled appointment could be actually discharged. Since monitoring in the IMC requires a highly qualified medical and nursing staff, the latest procedure would relieve the clinical-spatial and personnel-resources and the healthcare system. The aim should be saving resources in order to be able to maintain sufficient TRU and monitoring capacities in the future, which can lead to life-saving treatment.

## 6. Patents

There are no patents resulting from the work reported in this manuscript.

## Figures and Tables

**Figure 1 jcm-09-02516-f001:**
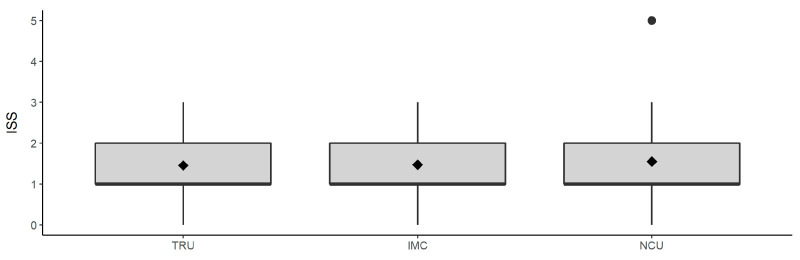
Distribution of ISS in each unit (TRU, IMC, NCU). Rhombuses represent the arithmetic mean. Medians are located at the bottom of the boxes (ISS = 1). The dot indicates outliers. TRU = Trauma Resuscitation Unit; IMC = Intermediate Care Unit; NCU = Normal Care Unit.

**Figure 2 jcm-09-02516-f002:**
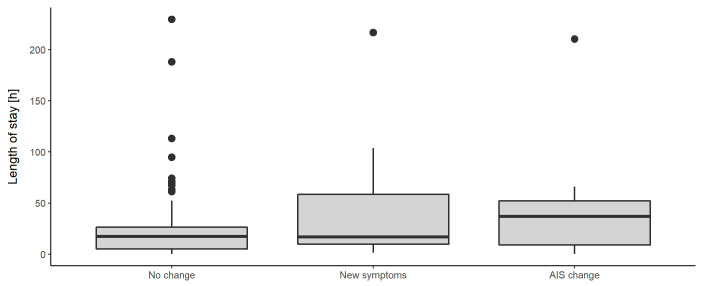
Distribution of length of stay (LOS) in correlation to the variables “No change“, “New symptoms“, and “AIS Change“. Dots represent outliers.

**Figure 3 jcm-09-02516-f003:**
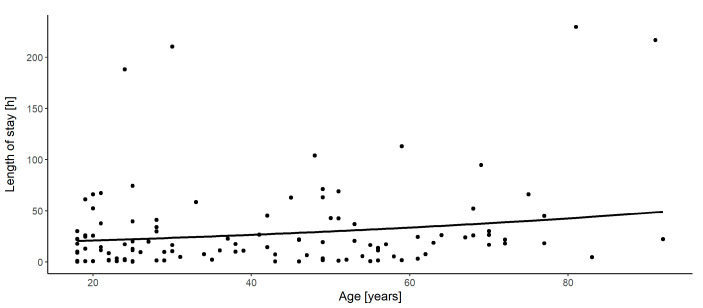
Correlation between length of stay and age of patients. The solid line (regression line) represents the correlation according to the transformed regression coefficient (Table 10).

**Table 1 jcm-09-02516-t001:** The severity of the Abbreviated Injury Scale (AIS).

	Injury Severity
1	Minor
2	Moderate
3	Serious
4	Severe
5	Critical
6	Maximum

**Table 2 jcm-09-02516-t002:** Demographic data.

N = 112		
**Age**	Mean = 41.18	SD = 19.63
**Gender n (%)**	69 (62%) male	43 (38%) female

N = Total sample size; n = Group sample size; SD = Standard Deviation.

**Table 3 jcm-09-02516-t003:** Mechanisms of injuries.

	Total (n)	Total (%)
**Motor vehicle accident**	62	55.36
**Motorcycle/bicycle accident**	9	8.04
**High and low falls**	20	17.86
**Pedestrian vs. vehicle crash**	12	10.71
**Physical abuse**	5	4.46
**Others**	4	3.57

n = Group sample size.

**Table 4 jcm-09-02516-t004:** ISS Average and MAIS.

	ISS Average *	Range **	MAIS *	Range **
**TRU**	1.45	0–3	1	0–1
**IMC**	1.47	0–3	1	0–1
**NCU**	1.55	0–5	2	0–2

TRU = Trauma Resuscitation Unit; IMC = Intermediate Care; NCU = Normal Care Unit; ISS = Injury Severity Score; AIS = Abbreviated Injury Scale; MAIS = Maximum Abbreviated Injury Scale. * Average of the ISS and AIS in the different units; ** Minimum and Maximum of ISS, AIS range.

**Table 5 jcm-09-02516-t005:** Frequencies of all observed ISS values in the units.

ISS	TRU	IMC	NCU
**0**	4	4	4
**1**	59	57	53
**2**	43	45	46
**3**	6	6	8
**5**	0	0	1

ISS = Injury Severity Score; TRU = Trauma Resuscitation Unit; IMC = Intermediate Care; NCU = Normal Care Unit.

**Table 6 jcm-09-02516-t006:** Frequencies of patients with new symptoms.

	N	% of N = 112
**Only IMC ***	9	8.036
**Only NCU ****	10	8.93
**Both**	2	1.78
**Total**	21	18.75

* IMC = Intermediate Care; ** NCU = Normal Care Unit; N = Total sample size; n = Group sample size.

**Table 7 jcm-09-02516-t007:** Comparison of patients’ frequencies with AIS and ISS ≥1 throughout stations.

				Number of N = 112 with Score > 0			Rate in % with Score > 0			RD in %	
**Score**	**Q**	**df**	**p** ****	TRU	IMC	NCU	TRU	IMC	NCU	TRU to IMC	IMC to NCU
**ISS ***	2	2	0.368	0	0	1	0.00	0.00	0.89	0.00	0.89
**AIS HN**	2	2	0.368	51	51	52	45.54	45.54	46.43	0.00	0.89
**AIS F**	3	2	0.223	14	15	16	12.50	13.39	14.29	0.89	0.89
**AIS C**	2	2	0.368	1	1	2	0.89	0.89	1.79	0.00	0.89
**AIS AbP**	2	2	0.368	0	0	1	0.00	0.00	0.89	0.00	0.89
**AIS ExP**	4	2	0.135	1	1	3	0.89	0.89	2.68	0.00	1.79
**AIS E**	3	2	0.223	96	97	98	85.71	86,61	87.50	0.89	0.89

* Binary coded (≤3 as 0 and >3 as 1); Q = Test value; df = Degrees of freedom; ** *p* = *p* value (Level of Significance = 0.05); RD = Risk difference (proportion in % at which the score changes); TRU = Trauma Resuscitation Unit; IMC = Intermediate Care Unit; NCU = Normal Care Unit; ISS = Injury severity score; AIS = Abbreviate injury scale; HN = Head or Neck; F = Face; C = Chest; Ab*p* = Abdominal or pelvic contents; Ex*p* = Extremities or pelvic girdle; E = External.

**Table 8 jcm-09-02516-t008:** Correlation between received diagnostic computed tomography (CT) and AIS change.

		AIS Alteration		Total
		No	Yes	
**No CT**	**Number**	7	0	7
	Expected number	6.4	0.6	7.0
	% within the category	100.0%	0.0%	100.0%
**Only CT body-trunk**	Number	4	0	4
	Expected number	3.7	0.3	4.0
	% within the category	100.0%	0.0%	100.0%
**Only CT head**	Number	24	1	25
	Expected number	23.0	2.0	25.0
	% within the category	96.0%	4.0%	100.0%
**CT head and body-trunk**	Number	68	8	76
	Expected number	69.9	6.1	76.0
	% within the category	89.5%	10.5%	100.0%
**Total**	Number	103	9	112
	Expected number	103.0	9.0	112.0
	% within the category	92.0%	8.0%	100.0%
**Alteration**	Chi^2^	df	*p* ***	
**AIS**	2.15	3	0.542	

Chi^2^ = Test Value; df = Degrees of freedom; * *p* = *p* value (Level of Significance = 0.05); CT = Computed tomography; AIS = Abbreviated Injury Scale.

**Table 9 jcm-09-02516-t009:** The average length of stay (LOS).

	Hours (%)
**Overall LOS**	53.57 (SD = 42.23; Median = 41.48)
**NCU**	34.65 (64.68%)
**IMC**	17.83 (33.28%)
**TRU**	1.08 (2.01%)
	Number (%)
**Patients who stayed >100 h**	6 (5.35%)
**Patients who stayed >200 h**	4 (3.57%)

LOS = Length of stay; NCU = Normal Care Unit; IMC = Intermediate Care Unit; TRU = Trauma Resuscitation Unit; SD = Standard Deviation.

**Table 10 jcm-09-02516-t010:** Regression analysis results for length of stay.

Variable	B	SE	*t*	*p* *	exp(B)
**(Intercept)**	2.81	0.30	9.36	<0.001	16.59
**Progress (Reference: No change)**					
**New symptoms**	−0.43	0.46	−0.94	0.350	0.65
**AIS change**	−0.16	0.55	−0.29	0.770	0.85
**Age (years)**	0.01	0.01	1.87	0.064	1.01
**Gender (Reference: male)**					
**Female**	−0.18	0.25	−0.71	0.481	0.84
**Extended diagnostics (Reference: none)**					
**Once**	1.28	0.56	2.31	0.023	3.61

* *p* = 0.05; B = Regression coefficient; SE = Standard error; t = Test value; * *p* = *p* value (Level of Significance = 0.05); exp(B) = Relative change.

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
