# Peer review of "Evidence of Prolonged Monitoring of Trauma Patients Admitted via Trauma Resuscitation Unit without Primary Proof of Severe Injuries"

_jcm, 2020, doi:10.3390/jcm9082516_

Round 1

Reviewer 1 Report

The authors present in this paper a retrospective evaluation on slightly injured patients admitted in a Trauma Resuscitation Unit (TRU). From the analysis of 112 patients, they reported that the AIS and ISS changes after initial evaluation in the TRU were null or minor, and therefore that monitoring in Intermediate Care Unit (IMC) should not be necessary, and consequently that those patients could be admitted directly after TRU into a normal care unit (NCU)

General comments:

The topic of this article is interesting, and the study seems to have been carefully performed. Regarding the manuscript, the text is clear and well presented. The tables and the figure are also clear and easy to read.

The only comment of the Reviewer is on the relevancy of this study : The intrahospital course TRU to IMC to NCU is based on the organisation of the hospital of the authors. In many hospital, slightly injured patients, after initial check-up in a TRU, are directly admitted in a NCU. For example, in my hospital, after initial check-up in the TRU, patients without any injury may go back directly to their home after 6 hours !  So the applicability of those results is only possible for hospital which have a similar organisation for care of trauma patients. 

Specific comments

The introduction is too long, with many “round trips” in the text between major trauma care and minor trauma care.

Author Response

Thank you very much for reviewing our manuscript and for your comments and suggestions.

We are glad to answer your comments:

General comments:

In Germany often – and the situation is similar in our centre – patients after TRU-management are admitted to an observation unit (IMC, ICU) though there is no significant injury found in the TRU. In hospitals where TRU-patients are dismissed after 6 hours they often are observed in the monitoring unit of the emergency area. We assume, the reason is in the german level 3 guideline on the treatment of patients with severe / multiple injuries, which induces a high rate of overtriage and gives explicit recommendations to the paramedics for activation the TRU. But there are no recommendations given for hospital treatment of patients with minor injuries. (“Traditionally” in Germany often cases with subjective uncertainties obtain a 24-h-observation “for safety”, e.g. minor skull traumas, electric shock with 220V. )

In Germany there is a habit of monitoring patients with minor injuries because of subjective uncertainties.Evidence for this behave can be found in the annual report of the “DGU-Traumaregister”, a nationwide database where most hospitals enroll trauma patients. Although an admission to ICU is compulsory to enroll a patient in that database 11 % of the registered patients had an MAIS of only 1. This underlines the current practice of a “better safe than sorry” policy without any evidence or official recommendation.

For this reason, it was the aim of our study to challenge a custom behaviour by providing scientific evidence.

Specific comment:

We shortened the introduction by more than 10 % and improved the terminology for better reading.

The new introduction reads as follows:

1. Introduction

Acute care of trauma patients in Germany, Austria, Switzerland, the Netherlands, Belgium and Luxembourg is ensured by local, regional and supraregional trauma centers (level I-III) according to the TraumaNetwork DGU® initiative.

TraumaNetwork DGU® initiative has enabled the German Society for Trauma Surgery (DGU) to establish first-class nationwide care for the severely injured. Clinics and University hospitals in order to provide high-quality patient care in trauma centers in addition to specialized medical requirements, also require specific spatial and material resources [1,2]. According to the guidelines of the DGU and the “Association of the Scientific Medical Societies in Germany” specific criteria must be fulfilled for the activation of the Trauma Resuscitation Unit (TRU). These criteria are divided into 3 groups: “disturbance of vital signs”, “detected injuries” and “mechanism of the accident or accident constellation” [2,3]. However, based on the last criterion it has been observed in Germany that many slightly injured patients are admitted to the TRU and only one in five is actually severly injured with an Injury Severity Score (ISS) ≥16 [4,5].

In 2019 the German Society for Trauma Surgery published the “Whitebook Medical Care of the Severely Injured” 3rd edition, which provides guidelines for the clinical and diagnostic steps, as well as for the first operative phase of the „Trauma Resuscitation Unit treatment” [2,6,7]. However until now there are no recommendations for the duration of treatment and whether further monitoring is necessary for the slightly injured trauma patients, who have suffered a dangerous trauma and who according to the TRU criteria must be treated in the TRU.

In addition, it has been shown that the intake of mind-altering substances (e.g drugs, alcohol or medication) in combination with a dangerous trauma, even if the patient is slightly injured, leads to the monitoring of the patient [8]. In individual and defined cases, if the patient has not suffered a severe injury, after implementation of the standardized Trauma Resuscitation Unit treatment, and if the occurrence of possible further complications is excluded, a prolonged monitoring appears not to be necessary [9–11]. However, relevant injuries such as cerebral contusion, occipital skull fracture or pneumothorax have been initially overlooked and were only subsequently detected [8].

This might be the reason, that it has been reported in the literature that many patients with minor trauma are admitted to the Intermediate Care Unit (IMC) for monitoring without any comprehensible medical reason [12].

A national or international study that considers the entire cohort of patients that have been treated in the TRU and were initially diagnosed as slightly injured does not exist.

In the present study it is therefore investigated whether inpatients surveillance after the interdisciplinary TRU diagnostic is necessary, so that injuries with an Abbreviated Injury Scale (AIS) ≥2 are not overlooked and secondly whether the inquired experience of the trauma team along with the improved computed tomography (CT) imaging makes the inpatient monitoring unnecessary.

We hope our remarks and addition have led to an improvement of the manuscript that meets your expectations. Thank you again for your efforts in supporting our work.

Reviewer 2 Report

Thank you for giving me the opportunity to review the manuscript. The authors concluded that an inpatient monitoring in the IMC for patients with an AIS ≤1 or an ISS ≤3 after completion of the TRU treatment is unnecessary. I agreed with the author's findings. While, I asked the authors to reply my concerns related to this study.

Major comment.

  1. Method

The authors included only the traumatized patients whose AIS<=1. In general. minor injury was thought to be AIS<=2. Why the authors did not include the patients whose AIS=2. I thought if the patients whose AIS=2 was included, the study's generalizability was increased.

    2. Result

AIS alteration among CT head and body-trunk patients was confirmed 8 patients. These 8 patients could not be found by whole-body CT? The injuries were all peripheral injuries?

    3. 112 patients had ISS <=3 and corresponded to 19.5% of the whole patients. I thought this number was high and the authors could discuss the rates compared with previous reports? 

Author Response

Thank you very much for reviewing our manuscript and for your comments and suggestions.

We are glad to answer your concerns.

  1. Method

We did not include patients with an AIS ≤2 for two reasons. First, the term “minor injury” is used for AIS = 1. AIS = 2 would be a “moderate injury”.

Aim of this study was to define a patient’s collective admitted via TRU which is not necessarily to be observed/monitored or hospitalized. We agree with you, that AIS = 2 is often (mostly) used for a rather “minor injury” and does not include dangerous diagnoses. But also there are a lot of injuries with an AIS = 2, which make an observation, hospitalization or operation necessary, e.g. Pneumomediastinum, vertebral body fracture without compression/dislocation or Subarachnoid hemorrhage.

So we think, that the group patients with AIS = 2 is to inhomogeneous for considering the necessity for hospitalization.

  1. Result

In a total of 9 patients an AIS change occurred during their hospital stay. All of these  9 patients  received a CT-scan of the head and 8 patients additionally a CT-scan of the body trunk. Thus a total of 8 patients with an AIS change received a trauma CT-scan.

The reasons why an AIS change occurred despite this comprehensive diagnosis are very different. Below is a brief explanation of the patients who, despite the use of a trauma CT-scan, experienced an AIS change during the course of the procedure.

Patient 1+2:

Two patients were diagnosed with new injuries to the extremities during the follow-up. In both cases the injuries were fractures of the extremities. In one case a fracture of the phalanx proximalis DIII of the right hand, and in the second case a basal limb fracture of the big toe of the right foot. The extremities far from the body trunk are not shown in the trauma CT-scan and thus were missed the initial diagnostics. Conventional imaging of the extremities was only carried out during the follow-up of newly occurring symptoms.

Patient 3:

In one patient there was a change of the AIS in the course of newly occurred pain in the area of a limb close to the body trunk, namely the thighs.. The thighs were shown in the initial trauma CT. Since a bone injury could be excluded, the complaints were classified as a contusion, and assigned to the soft tissue.

Patient 4:

One patient showed a restricted mouth opening and pain in the jaw. Here, too, a bony injury could be excluded in the trauma CT. Clinically it was a distortion of the temporomandibular joint and the new diagnosis was assigned accordingly.

Patient 5+6:

In two cases there were neurological complaints, one with tingling paresthesia in the legs and one with tingling paresthesia in the arms. In both cases bone injuries could be excluded by the trauma CT scan. In both patients the imaging was supplemented by an MRI. In one case an oedema in the lumbar spine was shown as a correlate of a distortion that had taken place. The MRI of the cervical spine of the second patient was inconspicuous. In both patients the new complaints were evaluated as a distortion of the respective spinal segment and assigned to the respective AIS region.

Patient 7:

In one case a posterior vitreous detachment was diagnosed only in the course of the procedure. The possible reasons are: 1. the initial imaging was unremarkable. This indicates that the findings in the TRU were not yet or not sufficiently prominent in the TRU to be recorded by CT diagnostics. 2. the additional CT was performed with another CT, which has a higher resolution.

Patient 8:

One patient was not diagnosed with a sternal fracture until the Tertiary Trauma Survey when the images were reviewed again. This fracture is retrospectively visible on the initial CT images. However, since the diagnosis was not seen in the first glance, it was not assigned to the admission diagnoses.

3.

We completed the discussion according to your suggestion and flaged the correspondend parts in the revised manuscript.

Our addition:

“Marzi showed an increase of patient admission to TRU of 70 % from 2012 to 2016 although the number of patients with an ISS ≥ 16 or an MAIS ≥ 3 is slightly constant over the years [4]. We made the similar observation in our centre and see this for the reason for the high number of patients with MAIS ≤ 1.”

  1. Marzi, I.; Lustenberger, T.; Störmann, P.; Mörs, K.; Wagner, N.; Wutzler, S. Increasing overhead ressources of the trauma room. Unfallchirurg 2019, 122, 53–58, doi:10.1007/s00113-018-0484-9.